

# Within-person variability in men's facial width-to-height ratio

Robin S.S. Kramer

Department of Psychology, University of York, York, United Kingdom

## ABSTRACT

**Background.** In recent years, researchers have investigated the relationship between facial width-to-height ratio (FWHR) and a variety of threat and dominance behaviours. The majority of methods involved measuring FWHR from 2D photographs of faces. However, individuals can vary dramatically in their appearance across images, which poses an obvious problem for reliable FWHR measurement.

**Methods.** I compared the effect sizes due to the differences between images taken with unconstrained camera parameters (Studies 1 and 2) or varied facial expressions (Study 3) to the effect size due to identity, i.e., the differences between people. In Study 1, images of Hollywood actors were collected from film screenshots, providing the least amount of experimental control. In Study 2, controlled photographs, which only varied in focal length and distance to camera, were analysed. In Study 3, images of different facial expressions, taken in controlled conditions, were measured.

**Results.** Analyses revealed that simply varying the focal length and distance between the camera and face had a relatively small effect on FWHR, and therefore may prove less of a problem if uncontrolled in study designs. In contrast, when all camera parameters (including the camera itself) are allowed to vary, the effect size due to identity was greater than the effect of image selection, but the ranking of the identities was significantly altered by the particular image used. Finally, I found significant changes to FWHR when people posed with four of seven emotional expressions in comparison with neutral, and the effect size due to expression was larger than differences due to identity.

**Discussion.** The results of these three studies demonstrate that even when head pose is limited to forward facing, changes to the camera parameters and a person's facial expression have sizable effects on FWHR measurement. Therefore, analysing images that fail to constrain some of these variables can lead to noisy and unreliable results, but also relationships caused by previously unconsidered confounds.

Corresponding author
Robin S.S. Kramer,
remarknibor@gmail.com

## INTRODUCTION

In the last decade, a great deal of research has focussed on one particular facial measure—width-to-height ratio (FWHR; *Weston, Friday & Liò, 2007*)—and its predictive power when considering a variety of human behaviours (for meta-analyses, see *Geniole et al., 2015*; *Haselhuhn, Ormiston & Wong, 2015*). Although originally proposed as evidence that sexual selection played a role in shaping the human skull (*Weston, Friday & Liò, 2007*), researchers have subsequently found associations between FWHR and aggression,

dominance, and threat behaviours in several domains (e.g., *Carré & McCormick, 2008*; *Stirrat & Perrett, 2010*; *Wong, Ormiston & Haselhuhn, 2011*). Interestingly, evidence suggests that FWHR is correlated with these behaviours, but it also predicts perceptions of faces when observers are asked to make judgements regarding these traits (e.g., *Carré, McCormick & Mondloch, 2009*; *Stirrat & Perrett, 2010*). As a result, it has been argued that FWHR is an evolved cue of threat (*Geniole et al., 2015*).

Although FWHR was originally measured directly from skulls (*Weston, Friday & Liò, 2007*), almost all studies linking this ratio with behaviours have collected measurements from 2D photographs (e.g., *Carré & McCormick, 2008*; *Stirrat & Perrett, 2010*). Evidence suggests that measurements taken from images show high agreement with measures taken directly from the face (*Kramer, Jones & Ward, 2012*), although the nature of this as a suitable proxy for skull FWHR has not been determined. More importantly for the current work, photographs of the same individual can vary dramatically (*Jenkins et al., 2011*). Unconstrained images of a face vary in pose, expression, lighting, age, camera settings, and so on. Such variability can significantly decrease face matching performance, i.e., telling if two different images are of the same person (e.g., *Megreya & Burton, 2006*; *Megreya & Burton, 2008*). Indeed, this within-person variability strongly argues against the idea that particular facial measures or distances underlie recognition (*Burton et al., 2015*).

If facial measures vary across images of the same person, is it reasonable to assume a reliable measure of FWHR can be obtained from a single 2D photograph? While lighting is unlikely to affect measures of the face (other than shadows preventing accurate measurement), several other variables may significantly alter a person's apparent FWHR. Previous research suggests that FWHR decreases with age (*Hehman, Leitner & Freeman, 2014*; cf. *Kramer, 2015*), although this is not generally controlled for in the literature (but see *Alrajih & Ward, 2014*). In addition, head pose (tilting upwards/downwards) has a sizable effect on FWHR obtained from photographs (*Hehman, Leitner & Gaertner, 2013*). This seems intuitive and, as a result, researchers have tended to include only images that are forward facing, i.e., looking directly at the camera without any noticeable tilting or left–right rotation.

In contrast, both facial expression and camera parameters appear less well considered. While many researchers have chosen to exclude images demonstrating expressions other than neutral (e.g., *Zilioli et al., 2015*), other researchers are less explicit in their inclusion criteria (*Haselhuhn & Wong, 2011*) or acknowledge that non-neutral images were included (*Carré & McCormick, 2008*). Regarding camera parameters, no FWHR research appears to have considered their effects. Interestingly, distance between the face and the camera, as well as the camera's focal length, are known to alter facial appearance (*Banks, Cooper & Piazza, 2014*; *Harper & Latto, 2001*; *Verhoff et al., 2008*), with those photographed closer to the camera appearing thinner and therefore having lower FWHRs (*Bryan, Perona & Adolphs, 2012*).

In previous studies, researchers have either failed to consider, or have simply avoided (through constraining photographic conditions), the potential influences of both facial expression and camera settings. Importantly, in many situations where images are collected from real-world contexts (e.g., political races, sporting competitions, etc.), no such

constraints can be imposed. In the current set of studies, I consider both influences on FWHR measurement. Through the calculation of effect sizes, I aim to determine how influential these factors might be, and hence whether researchers need to constrain or control for these effects in all future work.

To my knowledge, no previous research has included measurement of FWHR while systematically varying camera conditions or facial expressions. As such, it is difficult to make predictions regarding how these two factors may influence resulting measures. However, visual inspection of within-person photographic changes in facial expression suggests that these can produce significant alterations to FWHR. Therefore, I hypothesise that varying one's facial expression may have a larger effect on FWHR than differences between individuals. Similarly, with large changes to camera parameters (distance to camera in particular), we see noticeable FWHR differences (*Harper & Latto, 2001*). Again, I would predict that these camera effects may be larger than the effect on FWHR due to differences between people's faces.

In the studies that follow, I focus on within-person variability in white men only (see *Haselhuhn, Ormiston & Wong, 2015*). The majority of research has established links between FWHR and various aggressive or competitive behaviours in men, but has generally failed to find such relationships in women (e.g., *Carré & McCormick, 2008*; *Haselhuhn & Wong, 2011*; cf. *Lefevre et al., 2014*). In addition, there may be significant differences in FWHR across ethnicities (*Kramer, 2015*). For these reasons, I investigated the effects of facial expressions and camera parameters on FWHR measures in white men, while avoiding any noise due to differences between ethnicities, as the results would then be of the most relevance for the current literature.

## STUDY 1—UNCONSTRAINED CAMERA PARAMETERS

In this study, I investigated the influence of variability in camera parameters on resulting FWHR measures. Although all images were taken front-on and with a relatively neutral expression, variables including the camera used, its focal length, and its distance to the subject were unconstrained.

### Materials & methods

Five white male Hollywood actors were selected based on their ages and their prolific film appearances. For each actor, five films were chosen that were released while the actor was between the ages of 30 and 35 years. This limited age range minimised the possibility that age might influence any variability in FWHR both within and between actors (*Hehman, Leitner & Freeman, 2014*). However, this time period does allow for potential fluctuations in body weight, perhaps required for different roles, and this is known to influence FWHR (*Coetzee et al., 2010*; *Geniole et al., 2015*). For each film, five screenshots were taken using VLC Media Player in which the actor displayed a relatively neutral expression and was facing front-on to the camera (although gaze was often not directed at the camera). Each screenshot was taken from a different scene in the film, and no images included beards or glasses. As such, 25 images were collected for each actor.
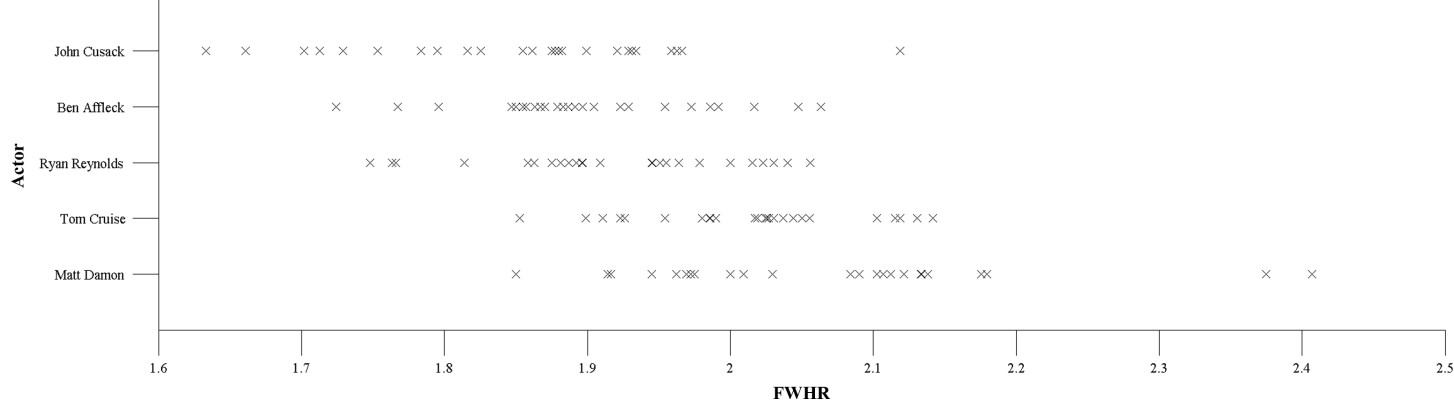

**Figure 1   A scatterplot illustrating the within-person variability in FWHR for each actor.** Each cross is a different image.

Following previous work (e.g., *Kramer, Jones & Ward, 2012*; *Stirrat & Perrett, 2010*), images were rotated using custom MATLAB software so that both pupils were aligned to the same transverse plane. The same software was then used to measure the width (the horizontal distance between the left and right zygions) and height (the vertical distance between the highest point of the upper lip and the highest point of the eyelids) of each image. The FWHR was calculated as width divided by height.

While every care was taken to include only images that were neutral and front-on, it must be acknowledged that there remained some variability in these two parameters, in particular where actors appeared relatively emotionally neutral but with their mouths open. As such, emotional expression could also be considered to vary here, although this is investigated more systematically in Study 3. Similarly, slight head rotations (both left/right and up/down) may also be present. Importantly, these images were still within the range of head angles that have been analysed in previous publications investigating real-world settings (e.g., *Carré & McCormick, 2008*; *Huh, Yi & Zhu, 2014*; *Kramer, 2015*; *Lewis, Lefevre & Bates, 2012*; *Loehr & O'Hara, 2013*; *Welker et al., 2014*; *Wong, Ormiston & Haselhuhn, 2011*), which are inherently less constrained than those taken in the laboratory (e.g., *Kramer, Jones & Ward, 2012*; *Özener, 2011*).

## Results

The variability in FWHR measures within and between actors can be seen in Fig. 1. It is clear that there is significant variability within actors regarding their FWHRs. As such, ordering these five actors in terms of FWHR would depend greatly on which particular image was considered to represent each man. The effect sizes due to Identity (the differences between actors) and Image (the differences within actors) can be quantified using sums of squares (SS) analyses (*Jones & Kramer, 2015*; *Morrison, Morris & Bard, 2013*). By dividing the SS for each factor (Identity, Image, Identity ×Image) by the total SS, I obtained their $\eta^2$ effect sizes. Although the five levels for the Identity variable made intuitive sense (each actor is a level), the 25 levels for Image were less meaningful since there is no relationship between the orders of images for each actor. Simply, there is no reason why Image 1 is the first image for John Cusack, and this bares no relationship with Image 1 for Ben Affleck. Therefore, in
order to obtain an idea of the effect sizes for the two factors and their interaction regardless of image orders, SS were calculated over 10,000 iterations, each time randomising the orders of images within actors. For Identity, the $\eta^2$ was 0.41 (and was unaffected by the ordering of the images within actors since this value only takes into account the differences between actors). For Image, the $\eta^2$ values over all iterations were $M = 0.12$, $SD = 0.03$, and for the Identity × Image interaction, the $\eta^2$ values were $M = 0.48$, $SD = 0.03$.

These analyses show that the differences between actors accounted for much more of the variance in FWHR than the differences within actors (due to image variation). This may seem surprising, and highlights the importance of identity differences, irrespective of the particular image chosen, when measuring FWHR. The largest effect size was due to the interaction between Identity and Image, suggesting that the differences between images depended on the actor, and were not equivalent across actors. This is to be expected, given the random selection of images—one actor's images may vary more than another's simply based on the particular images/films used.

I repeated this analysis using only one film (and therefore five images) per actor in order to better equate the variability in images within actors. With each actor limited to a shorter time period and a single role, the variability due to camera parameters remains while additional changes due to weight fluctuations and character changes are minimised. Only the first film in the set for each actor was considered, and effect sizes for Identity (0.54), Image ($M = 0.09$, $SD = 0.05$), and for the Identity × Image interaction ($M = 0.36$, $SD = 0.05$) mirrored the pattern seen above. However, Identity showed an even great effect here while the interaction effect decreased. By removing the changes in FWHR due to differences across films for a given actor, which perhaps have little equivalence in the real world, I found that FWHR was influenced more by differences between people than within (due to particular images).

Another way to quantify the importance of considering (and potentially constraining) camera parameters when selecting images is to model the rank correlation of the five actors irrespective of which image was used. This method more closely resembles analyses in the literature where FWHRs are correlated with behavioural measures, relying on the ordering (and specific values) of the faces. For each iteration, I randomly selected two images for each actor (from the 25 available). I then correlated the FWHRs for the five pairs of images, giving a measure of agreement between the rankings of the actors irrespective of which images were used. After 10,000 iterations, the distribution of rank correlations ($M = 0.41$, $SD = 0.42$) showed relatively low agreement. Even lower agreement was found when this analysis was repeated using only images from the first film for each actor ($M = 0.34$, $SD = 0.45$). Therefore, if researchers fail to constrain camera parameters during image collection, there will be a sizable effect on the orders of their actors according to FWHR.

## STUDY 2—VARIATION IN FOCAL LENGTH AND DISTANCE TO THE CAMERA

It is clear that camera parameters in relatively unconstrained images can have a significant influence on the apparent FWHR of a face. Next, I consider the effect of camera parameters

on FWHR under controlled laboratory conditions. By analysing images where focal length and camera distance were systematically varied, I can determine their particular influence on FWHR without additional noise due to head pose, emotional expression, etc.

## Materials & methods

Photographs of 21 white adult men were taken from the Caltech Multi-Distance Portraits database (*Burgos-Artizzu, Ronchi & Perona, 2014*). For each model, front-on images were taken using a Canon Rebel Xti DSLR camera at seven distances: 60, 90, 120, 180, 240, 360, and 480 cm. Longer focal lengths were used with greater distances in order to equate the sizes of the images. Models were instructed to remain still and maintain a neutral expression throughout the procedure, which lasted 15–20 s. No images included beards or glasses. As above, images were rotated using custom MATLAB software so that both pupils were aligned to the same transverse plane, and then FWHRs were measured.

## Results

As in Study 1, SS analyses were carried out on the FWHR values in order to quantify the effect sizes due to Identity (the differences between models), Distance (the differences due to the distance from the camera and its focal length), and their interaction. The results showed that Distance ($\eta^2 = 0.18$) had a much smaller effect on FWHR than Identity ($\eta^2 = 0.80$). This suggests that the particular camera distance–focal length combination had little effect on FWHR measures relative to the general differences between models. Even so, camera distance did have a statistically significant effect on FWHR, $F(6, 120) = 196.23$, $p < .001$, with FWHR increasing with greater distance to the camera (*Bryan, Perona & Adolphs, 2012*). Interestingly, the effect size of the Distance × Identity interaction was very small ($\eta^2 = 0.02$), suggesting that the way the camera distance and focal length altered FWHR was equivalent for all models.

In addition, I quantified the importance of considering (and potentially constraining) camera distance and focal length when selecting images by modelling the rank correlation of the 21 models irrespective of which image was used (see Study 1). After 10,000 iterations, the distribution of rank correlations ($M = 0.75$, $SD = 0.08$) showed high agreement, suggesting that if researchers fail to constrain camera distance and focal length, there will only be a limited effect on the orders or rankings of their models according to FWHR.

## STUDY 3—VARIATION IN FACIAL EXPRESSION

In addition to the influence of camera parameters on FWHR measurement, another source of within-person variability comes from facial expressions. The same face posing different expressions may significantly alter FWHR. In general, researchers utilise neutral expressions and exclude all others (e.g., *Welker et al., 2014*), but as yet, there has been no investigation into how expressions may systematically alter FWHR measurement.

## Materials & methods

Photographs of 20 white adult men were taken from the Radboud Faces Database (*Langner et al., 2010*). For each model, front-on images of eight emotional expressions (based on

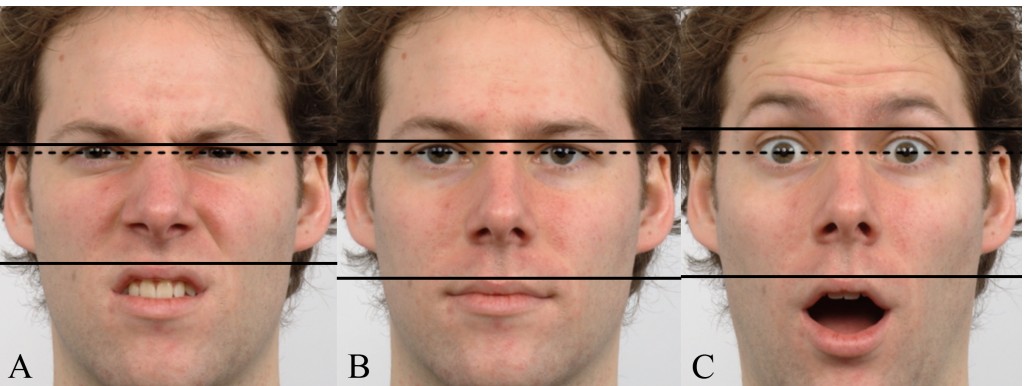

**Figure 2 Example images (after rotation) from the Radboud Faces Database (reproduced with permission).** Horizontal lines depict the facial height for disgust (A), neutral (B), and surprise (C) expressions. The dashed line illustrates that the pupils are level.

the Facial Action Coding System; *Ekman, Friesen & Hager, 2002*) were included: angry, contemptuous, disgusted, fearful, happy, neutral, sad, and surprised. No images included beards or glasses. As above, images were rotated using custom MATLAB software so that both pupils were aligned to the same transverse plane, and then FWHRs were measured. See Fig. 2 for examples.

## Results

As in Study 1, SS analyses were carried out on the FWHR values in order to quantify the effect sizes due to Identity (the differences between models), Expression (the differences due to posed expression), and their interaction. The results showed that Expression ($\eta^2 = 0.58$) had a much larger effect on FWHR than Identity ($\eta^2 = 0.31$). This suggests that the particular expression a person wears has a large influence on their FWHR, and can alter the rankings of a set of models. Interestingly, the interaction between these two factors was relatively small ($\eta^2 = 0.11$), suggesting that particular expressions alter the FWHRs of models in similar ways. For example, disgusted expressions may systematically increase FWHR measures across models while surprised expressions decrease them.

By carrying out a repeated measures ANOVA, treating Expression as an 8-level factor that varied within models, I was able to investigate which expressions significantly altered FWHRs in comparison with a baseline neutral expression. As expected, I found a significant effect of Expression, $F(7, 133) = 103.44$, $p < .001$. The results of pairwise comparisons between the neutral expression and the remaining seven expressions are illustrated in Fig. 3.

As Fig. 3 shows, models posing with disgusted or happy expressions significantly increased their FWHRs in comparison with neutral (both $ps < .001$). In contrast, posing with a fearful or surprised expression significantly lowered their FWHRs (both $ps < .001$). The remaining three expressions had no significant effect on FWHR (all $ps > .05$).

In addition, I quantified the importance of considering (and potentially constraining) facial expression when selecting images by modelling the rank correlation of the 20 models irrespective of which image was used. After 10,000 iterations, the distribution of rank
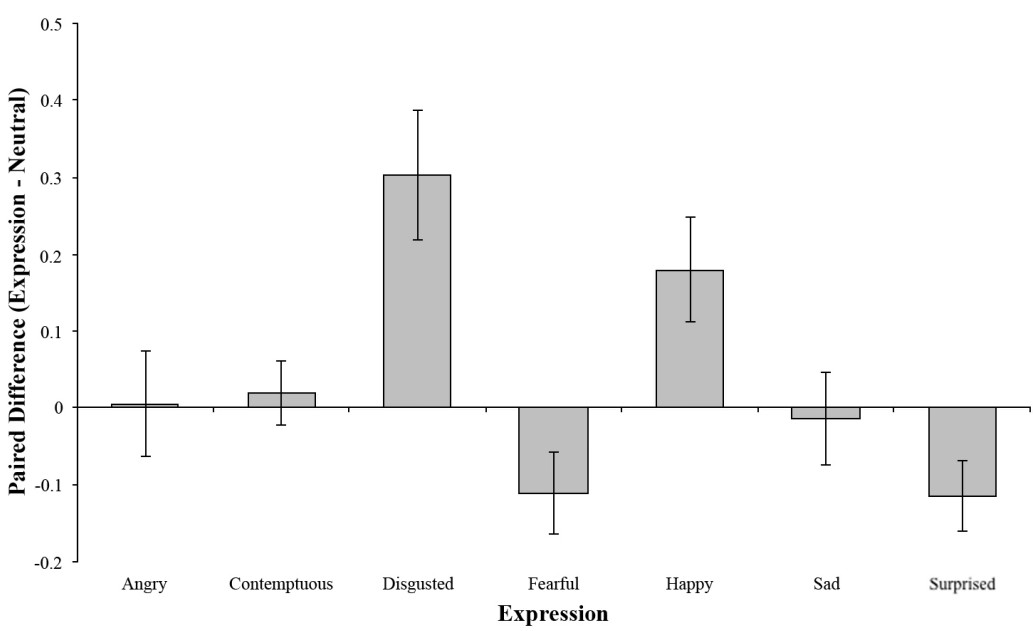

**Figure 3** **A bar chart illustrating the pairwise comparisons between the neutral expression and the remaining seven expressions.** The error bars represent the 95% confidence intervals of the differences, adjusted for multiple comparisons using Bonferroni correction. As such, error bars overlapping the zero line show no difference from neutral.

correlations ($M = 0.23$, $SD = 0.20$) suggests that if researchers fail to constrain expression during image selection, there will be a sizable effect on the orders of their models according to FWHR.

## DISCUSSION

Across three studies, I investigated the influences of camera parameters and facial expressions on FWHR measurement. While within-person variability can have a substantial effect on FWHR measurement, this was not always the case.

The results of Study 1 suggest that failing to constrain camera parameters, or indeed the camera used, may not be as detrimental to a study's design as one might predict. Differences between identities accounted for a larger proportion of the variation in FWHR than differences within identities (across images). This identity effect became larger still, relative to within-person image differences, when images were limited to only one film per actor, which might be considered more comparable to the variation one might expect in everyday faces. However, correlation analyses highlighted the substantial effect within-person differences could have on the ranking or ordering of faces, an important issue for the majority of articles on this topic. Therefore, collecting images taken by different cameras using different settings (in contrast with the more constrained parameters of Study 2) will add substantial noise to any potential FWHR–behaviour relationship, or on occasion, may even lead to the detection of spurious relationships if these factors are confounded with the variables under investigation. For example, if I were to compare the FWHRs of Democratic and Republican candidates in the US, and these two political parties utilised

two different photographers, it may be that the differing camera set-ups result in apparent FWHR differences. Indeed, two sets of white men of approximately the same age, taken using different camera set-ups, produced significantly different FWHRs in previous work (Study 1 vs. Study 2 in *Kramer, Jones & Ward, 2012*).

Here, only five actors were included in Study 1 and so the specific effect sizes may alter with a larger sample. Of course, one could also collect more images for each actor. The important result is not the values themselves but the relative sizes of the effects. Indeed, the FWHR values illustrated in Fig. 1 are comparable with those obtained in previous studies (e.g., *Kramer, Jones & Ward, 2012*; *Özener, 2011*). It is clear that even when relatively neutral, front-on images are used, there can be large variation in FWHR for a single face.

While every care was taken to constrain the images used in Study 1, slight head rotations and expressions may have been present. For example, it can be difficult to detect, and therefore control, up/down head tilt in two-dimensional images. Previous research has demonstrated that people may tilt their heads in order to affect perceived FWHR and appear more intimidating (*Hehman, Leitner & Gaertner, 2013*). In order to control for the noise in FWHR measurement due to head position, researchers might utilise three-dimensional imaging where possible (e.g., *Kramer, Jones & Ward, 2012*).

Study 2 showed that the distance to camera itself, along with alterations to focal length, have only a relatively small effect on FWHR in comparison with between-subject differences. This was also demonstrated when I considered the ranking or ordering of identities by FWHR. Therefore, these factors appear less important when compared with the more unconstrained images used in Study 1. Unfortunately, because camera distance and focal length were both allowed to vary in the particular image set used, further research is needed in order to separate out the influences of these two parameters.

Interestingly, I found only a small effect of the interaction between camera parameters and identity in Study 2, suggesting that increasing the camera distance and focal length alters FWHR consistently across different people. Indeed, this result has meant that computer scientists have found some success in estimating camera distance using photographs of unknown people (*Burgos-Artizzu, Ronchi & Perona, 2014*; *Flores et al., 2013*).

In Study 3, I found that FWHR changed substantially across different expressions. Therefore, as researchers have already implicitly assumed, it is important to keep this variable constrained when collecting image sets. However, the current results also suggest a caveat—only four of the seven expressions investigated here significantly differed from neutral. As a result, angry, contemptuous, and sad facial expressions may not require exclusion during image collection (assuming the majority of images are neutrals). Importantly, happy expressions (i.e., smiles) produced significantly larger FWHRs and these are the expressions that tend to appear most in photographic sets (given that people often smile unless instructed otherwise). Therefore, inclusion of these images may lead to potentially spurious results. For example, the recent controversy surrounding sexual dimorphism in FWHR (e.g., *Kramer, Jones & Ward, 2012*; *Özener, 2011*) could be unintentionally affected if expression was not tightly constrained during photography, given that women tend to smile more than men in various situations (*LaFrance, Hecht & Paluck, 2003*). Interestingly, smiling faces are also perceived as more competitive (*Mehu,*

*Little & Dunbar, 2008*), which fits well with research whereby faces with larger FWHRs are perceived as more dominant, aggressive, etc. (e.g., *Stirrat & Perrett, 2010*).

Facial height in the current work was measured as the distance from the highest point of the upper lip to the highest point of the eyelids (*Kramer, 2015*; *Kramer, Jones & Ward, 2012*; *Stirrat & Perrett, 2010*). It is worth mentioning that other researchers have instead chosen to use the brow as the upper boundary (e.g., *Carré & McCormick, 2008*; *Özener, 2011*). It may be that facial expressions have an even larger effect on FWHR measures if the brow is used, given the sizable shift in the position of the eyebrows for a number of expressions (*Ekman, Friesen & Hager, 2002*).

The results presented here are derived from white male faces only. As discussed in the Introduction, the majority of findings to date regarding FWHR and its associations with behaviours are in white men. However, there is no *a priori* reason to believe camera manipulations or changes to facial expression have different sizes of effects in women or other ethnicities. Of course, expressions may alter women's faces in systematically different ways because their face shape, and hence the way they pose expressions, differs from men. As such, I invite future researchers to address this question.

The current work focusses on potential issues when measuring FWHR from photographs, given its variability across different images of the same individual. These issues are also relevant when considering the broader topic of social trait inferences. Recently, researchers have been investigating how different images of the same face can produce widely varying social perceptions of that individual (*Jenkins et al., 2011*). Foreseeing the work presented here, *Todorov & Porter (2014)* noted that even invariant facial characteristics like FWHR are susceptible to image variation. Combining these topics, there is evidence to suggest that inferences more likely to be based on static cues (like FWHR) may vary less across images of the same face in comparison with inferences based more on dynamic cues like muscle movements (*Hehman, Flake & Freeman, 2015*). This growing body of research demonstrates how image variation can have repercussions for several areas of study.

## CONCLUSIONS

The current set of studies explores the importance of considering both camera parameters and facial expressions when investigating FWHR. With increasing numbers of researchers downloading images from the Internet in order to explore real-world contexts (e.g., politicians, presidents, professional fighters, football players), the ability to control these factors may be lost. Critically, one must then question whether it is even meaningful to compare images where the camera set-up varies across individuals, for example. To date, there has been no experimental consideration of this particular factor to my knowledge. In conclusion, I recommend that future researchers consider whether both camera parameters and facial expressions can be constrained, and indeed need to be constrained, during image collection before undertaking real-world investigations.

## ACKNOWLEDGEMENTS

I thank Alex Jones and Renée Lefebvre for comments on the manuscript.

### Funding

The author received no funding for this work.

### Competing Interests

The author declares that there are no competing interests.

### Author Contributions

- Robin S.S. Kramer conceived and designed the experiments, performed the experiments, analyzed the data, contributed reagents/materials/analysis tools, wrote the paper, prepared figures and/or tables, reviewed drafts of the paper.

### Data Availability

The data set is available in the Supplemental Information as Table S1.

### Supplemental Information

Supplemental information for this article can be found online at http://dx.doi.org/10.7717/peerj.1801#supplemental-information.

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
