# Peer review of "Within-person variability in men’s facial width-to-height ratio"

_PeerJ, doi:10.7717/peerj.1801_

## Round 0.1 · original submission · Minor Revisions

· Academic Editor

Minor Revisions

Like me, the reviewers are impressed with the quality of your manuscript. I look forward to accepting it once you attend to their suggestions. Thank for your submission.

Reviewer 1 ·

Basic reporting

The paper is well written and acknowledges the relevant background literature.

Here are a few minor edits that I suggest:

Line 101-102: the author states that the majority of work in fwhr has looked at men, which is correct. However, it would be good to acknowledge those studies that have shown links in women. I am aware of one such study: Lefevre, C. E., Etchells, P. J., Howell, E. C., Clark, A. P., & Penton-Voak, I. S. (2014). Facial width-to-height ratio predicts self-reported dominance and aggression in males and females, but a measure of masculinity does not. Biology letters, 10(10), 20140729.

Line 123: the first paper that showed links between fwhr and BMI to my knowledge was Coetzee et al. (Coetzee, V., Chen, J., Perrett, D. I., & Stephen, I. D. (2010). Deciphering faces: Quantifiable visual cues to weight. Perception, 39(1), 51-61.) I suggest citing this paper here.

Experimental design

The experimental design appears appropriate.

Validity of the findings

This is mostly fine. I have one comment about the discussion section:

In lines 289-296 the author states that the results of study 1 may suggest that systematic variation in camera set-up is problematic and needs to be taken into account. However, in my reading of the manuscript, study 2 directly contradicts this claim, showing clearly that focal length and camera distance don't have great effects on fwhr. As such, I suggest rewording or deleting this passage as it may confuse the reader.

The author does not mention head tilt in the discussion, I wondered whether differences in head tilt relative to the camera may be a major source of fwhr variation, despite him choosing only images that were face-on. up and down tilt can be hard to detect in a 2d image. It would be good if the author could comment on this, perhaps with the view to using more 3d images in future, which has been the case for some studies in fwhr.

Reviewer 2 ·

Basic reporting

This is a great paper. It is well written and addresses the limitations of a widely used measure in psychology. I have only one suggestion. The paper's relevance can be increased if the issues discussed here are related to broader issues in psychology: how image variation affects personality inferences. The author cites the seminal work of Burton and Jenkins who introduced this problem. There have been 2 recent reports that extended their work to personality inferences from face images. The first is Todorov & Porter (Psychological Science, 2014, 25, 1404). These authors actually argued that measures like the WHR are not invariant with respect to image variation (p. 1415), but there has not been any data with respect to this argument. Until the present paper, that is. The second relevant paper is Hehman et al. (Personality and Social Psychology Bulletin, 2015, 41, 1123). Incidentally, these authors used the WHR measure in their research. And the conclusions might be a bit different but the general argument is in the same spirit, namely that incidental image variation cam generate very different personality impressions.

Experimental design

Sound.

Validity of the findings

Good. The three experiments are quite different but with convergent findings.

Additional comments

My suggestion is to include one general paragraph in the discussion situating this work in a broader context.

---

## Round 0.2 · accepted · Accept

· Academic Editor

Accept

Thanks for a speedy and effective revision. I'm confident your paper will make an important contribution to this field.